# Potential Role of Exosomes in the Chemoresistance to Gemcitabine and Nab-Paclitaxel in Pancreatic Cancer

**DOI:** 10.3390/diagnostics12020286

**Published:** 2022-01-23

**Authors:** Annalisa Comandatore, Benoit Immordino, Rita Balsano, Mjriam Capula, Ingrid Garajovà, Joseph Ciccolini, Elisa Giovannetti, Luca Morelli

**Affiliations:** 1General Surgery Unit, Department of Translational Research and New Technologies in Medicine and Surgery, University of Pisa, 56124 Pisa, Italy; a.comandatore@libero.it; 2Department of Medical Oncology, Cancer Center Amsterdam, Amsterdam UMC, Vrije Universiteit Amsterdam, 1081 HV Amsterdam, The Netherlands; rita.balsano@gmail.com; 3Fondazione Pisana per La Scienza, 56124 Pisa, Italy; benoit-immordino@hotmail.com (B.I.); m.capula@fpscience.it (M.C.); 4SMARTc Unit, Centre de Recherche en Cancérologie de Marseille, Inserm U1068 Aix Marseille Université, 13385 Marseille, France; joseph.ciccolini@univ-amu.fr; 5Medical Oncology Unit, University Hospital of Parma, 43100 Parma, Italy; i.garajova@gmail.com

**Keywords:** exosomes, pancreatic cancer, chemoresistance, gemcitabine, nab-paclitaxel

## Abstract

In recent years, a growing number of studies have evaluated the role of exosomes in pancreatic ductal adenocarcinoma cancer (PDAC) demonstrating their involvement in a multitude of pathways, including the induction of chemoresistance. The aim of this review is to present an overview of the current knowledge on the role of exosomes in the resistance to gemcitabine and nab-paclitaxel, which are two of the most commonly used drugs for the treatment of PDAC patients. Exosomes are vesicular cargos that transport multiple miRNAs, mRNAs and proteins from one cell to another cell and some of these factors can influence specific determinants of gemcitabine activity, such as the nucleoside transporter hENT1, or multidrug resistance proteins involved in the resistance to paclitaxel. Additional mechanisms underlying exosome-mediated resistance include the modulation of apoptotic pathways, cellular metabolism, or the modulation of oncogenic miRNA, such as miR-21 and miR-155. The current status of studies on circulating exosomal miRNA and their possible role as biomarkers are also discussed. Finally, we integrated the preclinical data with emerging clinical evidence, showing how the study of exosomes could help to predict the resistance of individual tumors, and guide the clinicians in the selection of innovative therapeutic strategies to overcome drug resistance.

## 1. Introduction

As of today, pancreatic ductal adenocarcinoma (PDAC) remains a highly lethal disease, with a 5 year survival rate for under 3% in patients with advanced disease upon diagnosis [1,2]. This dismal prognosis reflects the aggressive metastatic behavior of PDAC and its remarkable chemoresistance driven by intrinsic and extrinsic factors related to the cancer cells [3,4].

PDAC is indeed a hard-to-treat tumor that has received limited attention and underinvestment from a research perspective [5], despite the fact that it is common (fourth most common major cancer and the incidence is predicted to increase in the next decade) and overall survival has not significantly improved in several decades [6,7]. Currently only about 15% of all pancreatic cancer patients can have surgery, which has been seen as the standard radical treatment. However, this treatment is still associated with significant morbidity, and increasing data are driving a shift from up-front surgery to neoadjuvant treatments within multidisciplinary therapeutic strategies of even resectable PDAC [8].

A signature hallmark of PDAC is its high degree of resistance against virtually any kind of therapy. In particular, the success of the most commonly used first-line chemotherapy regimens, such as gemcitabine+nab-paclitaxel [9,10] is strongly limited by primary and/or acquired chemoresistance, and an improved understanding of its underlying mechanisms is essential to improve the overall prognosis of PDAC.

Interestingly, in the last decade there has been an exponential rise in the number of scientific studies about the role of tumor-derived extra-cellular vesicles (EVs, also known as exosomes) in the cancer pathophysiology and several studies have shown that exosomes from PDAC cells or other cells in the tumor microenvironment (TME) can promote chemoresistance by modulating several target genes, RNAs, proteins, and oncogenic pathways which affect the therapeutic activity of anticancer drugs used in the clinical management of PDAC [11,12].

Therefore, the present review gives an update on the current literature of PDAC chemoresistance, with a special focus on the potential role of exosomes in reducing the activity of gemcitabine and (nab)-paclitaxel (Figure 1).

## 2. Pancreatic Cancer Chemoresistance

Cancer chemoresistance has historically been associated with genetic factors. The increased knowledge of PDAC genomics, including large-scale studies defining specific omics subtypes has provided insights into key mechanisms of pathogenesis. However, this has not yet been translated to the identification of “actionable” therapeutic targets, or clinically relevant approaches to improving patient care beyond a subgroup of *BRCA1-2* or *DNA-repair* genes mutations [13].

The dense desmoplastic mass, including cancer-associated fibroblasts (CAFs), immunosuppressive TME, and extracellular matrix stiffening [14] is another hallmark of PDAC and means that drugs are unlikely to be properly distributed at the tumor site. Therefore, PDAC TME has frequently been reported as a major contributor to chemoresistance. Approximately 80% of the PDAC mass is indeed constituted by stroma comprising a cellular component (stellate cells, fibroblasts, endothelial, and immune cells), an extracellular matrix, and a liquid milieu of cytokines/growth factors and exosomes. All these components are interconnected and their communication with cancer cells might affect aggressive behavior and therapy response [15]. In addition, PDAC stroma harbors inflammatory cells that might suppress cancer-directed immune mechanisms and contribute to hypoperfusion, reducing drug delivery and activity, as well as immune evasion. This led to the development of anti-stromal treatments, which have largely failed, as well as immunotherapeutic approaches [16,17].

These unsatisfactory clinical results suggest that important additional factors influencing PDAC chemoresistance are still to be fully elucidated, and new research is warranted to extend the limited list of current biomarkers with potential clinical application (Appendix A).

Recent studies support the role of PDAC-derived exosomes in preconditioning the liver or lung to render them vulnerable to metastatic cell engraftment [18,19]. Remarkably, exosomes released by malignant breast cells can be taken up by less malignant cells also in distant tumors, and carry RNA involved in metastasis [20]. These findings are prompting studies on therapies targeting critical steps for exosomes’ activity, such as biogenesis, release, cell-uptake, or downstream signaling. However, the potential role for tumor exosomes as the “fuel supply-line” for metastasis also offers an important opportunity to redirect the focus from the cancer cell and local microenvironment to a wider tumor crosstalk with metastatic foci that could be exploited in order to overcome PDAC chemoresistance.

## 3. Mechanisms Underlying Resistance to the Gemcitabine plus Nab-Paclitaxel Regimen 

Gemcitabine plus nanoparticle albumin-bound nab-paclitaxel represents one of the standards of care in advanced PDAC therapy and is suitable for a broader spectrum of patients compared to other schedules, such as the more toxic FOLFIRINOX regimen [21]. Moreover, the selection of gemcitabine plus nab-paclitaxel as a first-line regimen allows the use of a non-cross-resistant second-line chemotherapy with both oxaliplatin and nano-liposomal irinotecan, whereas FOLFIRINOX does not.

In the phase III MPACT trial, gemcitabine plus nab-paclitaxel was associated with a significantly higher objective response rate (23%) and a significantly longer median overall (OS, 8.5 months) and progression-free survival (PFS, 5.5 months) in comparison to gemcitabine monotherapy [22]. 

It was hypothesized that the albumin-bound nab-paclitaxel could selectively accumulate in the PDAC via the secreted protein acidic and rich in cysteine (SPARC) matricellular glycoprotein which binds albumin and is overexpressed in PDAC stroma [23]. High expression levels of SPARC were indeed correlated to poor survival outcomes in the phase II trial, suggesting its use as a possible predictive biomarker for nab-paclitaxel [24]. However, these findings were not confirmed in the phase III trial and Neesse and collaborators demonstrated that SPARC expression in the stroma did not correlate with drug accumulation in an in vivo model of PDAC [25].

Of note, preclinical studies showed that the combination of gemcitabine and nab-paclitaxel increased intratumoral gemcitabine levels due to a relevant decrease in cytidine deaminase (CDA), the main enzyme involved in gemcitabine catabolism. In agreement with these findings, paclitaxel reduced the levels of CDA through ROS species-mediated degradation, resulting in the increased stabilization and activity of gemcitabine in PDAC cells [26]. 

Despite the superiority of gemcitabine plus nab-paclitaxel versus gemcitabine alone in both OS and PFS [27], chemoresistance has a strict impact on the effectiveness of this regimen. However, the mechanisms underlying this resistance are only partially known. The activity of gemcitabine depends on several steps: first, intracellular uptake is necessary, and it is mediated by a nucleoside transporter and mainly by hENT1 [28]. Reduced expression of hENT1 can cause resistance to gemcitabine, whereas overexpression is correlated with prolonged OS [29]. Of note, a recent study showed that hENT1 expression correlated with PFS also in patients treated with the gemcitabine plus nab-paclitaxel regimen enrolled in the Comprehensive Molecular Characterization of Advanced Ductal Pancreas Adenocarcinoma for Better Treatment Selection (COMPASS) trial [30].

After intracellular transport, gemcitabine is phosphorylated by deoxycytidine kinase (dCK) [31]. The inactivation of this enzyme limits the cytotoxic activity of the drug [32] while higher levels of dCK leads to restored sensitivity [33]. Ribonucleotide reductase (RR) is another key determinant of gemcitabine activity. This enzyme is involved in the synthesis of DNA, and is inhibited by gemcitabine 5'-diphosphate (dFdCDP), an active nucleoside mediating gemcitabine cytotoxicity [34]. Overexpression of RRM1, one of the subunits of RR, has been observed in patients receiving gemcitabine and was associated with shorter survival in a recent meta-analysis, representing another potential mechanism of chemoresistance in PDAC patients treated with gemcitabine [35]. 

Studies regarding chemoresistance to nab-paclitaxel in PDAC are more limited, but the similarity with paclitaxel and its use in many types of cancer suggest several possible mechanisms mediating resistance, including EMT [36]. Three major proteins, namely P-glycoprotein (P-gp, also known as multidrug resistance protein 1 (MDR1) or ATP-binding cassette sub-family B member 1 (ABCB1)), MDR-associated protein 1 (MRP1), and breast cancer resistance protein (BCRP), which play a critical role in multidrug resistance, are indeed involved in resistance to all the taxanes [37]. Considering the effect of paclitaxel on microtubules, resistant cancer cells can also develop alterations of the β-tubulin family [29]. In particular, in PDAC, β-tubulin III overexpression has been related to enhanced tumor growth and metastasis, supporting new strategies silencing this specific isotype [38]. Another gene that seems to have a specific role in the chemoresistance of PDAC is *human epididymis protein 4* that causes both cancer cell growth and reduced sensitivity to paclitaxel in PDAC cell lines. These results suggest that HE4 expression levels may be used to predict the sensitivity of PDAC patients to paclitaxel [39]. 

More recently, Parasido and collaborators studied resistance to nab-paclitaxel in primary cultures as well as in zebrafish and in athymic nude mouse xenograft models. Interestingly, this study found that the sustained induction of c-MYC was associated with resistance while its depletion or treatment with either the MEK inhibitor, trametinib, or a small-molecule activator of protein phosphatase 2a restored chemosensitivity [40]. However, further studies are necessary to investigate chemoresistance and studies of the combination regimen are urgently needed to better understand how to overcome resistance and how it can impact positively on the survival of PDAC patients.

## 4. Exosomes

By definition, exosomes are a heterogeneous population of extracellular membranous vesicles with a diameter ranging from 40 to 100 nm that carr various cargo such as lipids, proteins, and nucleic acids that are important for cell-to-cell communication [41].

Exosomes are derived from the endocytotic system. Firstly, early endosomes are formed from the plasma cell by encapsulating the intracellular molecules as proteins and nucleic acids. After the maturation of the endosomes, exosomes are secreted after the fusion of multivesicular body (MVB) with the plasma membrane [42]. Exosomes are secreted by cells of all tissues and organs in both healthy and diseased conditions and their molecular content can change according to the donor cell type [43].

Exosomes can deliver information to recipient cells by releasing their cargo into the cytosol through direct fusion with the plasma membrane or may be internalized through several endocytic mechanisms (Figure 2). Alternatively, they can also act at the cell surface, without the need to deliver their contents [44]. Most importantly, the transferred content from exosomes (e.g., mRNAs, miRNAs, or proteins,) is functional, and can regulate the functions and of the recipient cells [20].

Exosomes can be extracted from serum, saliva, plasma, urine, and from cell cultures, using cell specific markers and immunoassays that enable their cell origin to be determined [45]. This feature makes the exosomes a particularly appealing source of new cancer biomarkers as well as for studies to investigate cancer progression. 

In particular, the content of exosomes released into the systemic circulation and/or bodily fluids has been increasingly explored as a potential rich source of non-invasive diagnostic and prognostic markers, which might be used as a surrogate for tumor biopsies, allowing early diagnosis, prognostication, and for predicting cancer progression as well as responses to therapy. A major breakthrough has been the discovery of cell type-specific proteins in PDAC cell-derived exosomes, as shown in a study where the serum glypican-1 positive+ exosome level detected early-stage pancreatic cancer with a sensitivity and specificity of 100% [46]. However, later studies have suggested that glypican-1 alone has a sensitivity of 82% and a specificity of 52% [47,48], and there seems to be no clinical adaptation of these exosome-assays, despite the strong association of glypicans with PDAC [49]. One possible explanation is that in these early studies the preanalytical conditions of plasma sampling as well as the exosome isolation procedures were suboptimal. Indeed, when studying secreted PD-L1 as a biomarker for an immunotherapy response prediction it became clear that bonafide exosome-association is a critical component [50]. The expression of specific tissue-derived and serum exosomal miRNAs such as miR-192-5p or miR181a has also been associated with the ability to distinguish PDAC from healthy control patients [51]. In one of the largest clinical studies on exosomes, Yu and collaborators [48] described a diagnostic (d-) signature for the detection of PDAC based on plasma RNA exosomes profiling of 501 patients. Hopefully this signature will be validated in multicenter studies and could be used for improving the early diagnosis and stratification of PDAC. However, a critical assessment of a number of analytical variables and the use of standardized techniques is still warranted in order to exploit the potential of exosomes-based “liquid biopsies”.

Other recent studies have underlined the role of exosomes in tumor progression, even if the mechanisms leading to aberrant exosome production in carcinogenesis remain largely unknown. For instance, modifications in stromal and tumoral syndecans lead to exosomal production and turnover, and contribute to PDAC progression and aggressiveness [52]. Some growth factors can contribute to cancer development affecting exosomal biogenesis as well [53]. Additional studies have identified specific proteins implicated in exosomes biogenesis, such as members of the integrin pathway, G-protein-coupled receptors (GPCRs), members of Notch signaling, and cytokines. These proteins can interact with proteins co-localized in the exosomes such as Tspan8 and CD44v6 during exosomal formation [54]. However, growth factors such as EGFR also take part in the uptake of exosomes targeting tumor cells. In addition, growth factors help exosomes in the release of intracellular information across the TME [52], and several oncoproteins such as KRas play a role in the exosomal pathway [55,56], suggesting there are many pathways and factors which regulate exosome release and the following effects during tumorigenesis.

Interestingly, recent studies have focused their attention on the role of exosomes in the immune response in the PDAC stroma [57]. Indeed, signals released by exosomes can modify cell composition in PDAC TME, changing immunosuppressive in immune effector cells [52]. Consequently, inhibiting the release or uptake of exosomes could be considered as a new strategy to improve immunotherapy approaches. TME reprogramming has also been demonstrated in a few studies showing how exosomes, either through exosomal miRNA, exosomal proteins, or signaling factors could contribute to PDAC metastasis [18,19,52].

Beyond all these features, exosomes could contribute to PDAC chemoresistance, and the following paragraphs will focus on this remarkable aspect, presenting an overview of the current knowledge about the role of exosomes in resistance towards gemcitabine and (nab)-paclitaxel.

## 5. Role of Exosomes in Resistance to Gemcitabine in Pancreatic Cancer

Gemcitabine has been the most used first-line drug for the treatment of PDAC over the last 25 years, and is currently approved both as monotherapy and in combination with nab-paclitaxel [58,59]. However, the efficacy of gemcitabine is limited by both primary and acquired chemoresistance which can be mediated by multiple mechanisms, as reviewed previously [60]. 

Here we report the main molecular mechanisms used by exosomes to transfer gemcitabine resistance competences to cancer cells, as summarized in Table 1.

### 5.1. Modulation of the Equilibrative Necleoside Transporter 1

A major role in the uptake of gemcitabine is played by the human equilibrative nucleoside transporter 1 (hENT1) [61]. Multiple studies, including analyses within the RTOG9704 and ESPAC-3 trials have evaluated hENT1 expression as a biomarker for gemcitabine efficacy in PDAC [62,63], showing that high hENT1 levels were associated with a significantly longer OS in patients receiving gemcitabine [61]. Of note, the association between high hENT1 expression and survival was also reported in patients treated with gemcitabine and nab-paclitaxel within the COMPASS trial [30].

Several molecular factors might influence hENT1 expression and the improved knowledge of such factors should help the identification of subgroups of patients who may benefit from gemcitabine-based therapies and overcome chemoresistance. For instance, the analysis of a collection of databases of miRNA–gene interactions (through the multimir R package) found a total of 175 miRNAs that potentially targeted hENT1 [61]. Interestingly, four of these are miRNAs are overexpressed in PDAC and one of these miRNAs, MiR-196a-3p, is upregulated in the exosomes of PDAC cell lines and in the exosomes in the serum of PDAC patients (at localized stages, i.e., I and IIa). However, data on responses to gemcitabine or about the outcomes are missing and further studies are needed to validate the hypothesis that this miRNA could be used as a surrogate biomarker of hENT1 expression and predict gemcitabine chemoresistance.

### 5.2. Modulation of Multidrug Resistance Proteins

The role of multidrug resistance (MDR) proteins in gemcitabine resistance is controversial. Indeed, a recent study has suggested that in gemcitabine-resistant PDAC cells, *MRP5* is expressed at higher levels than gemcitabine-sensitive cells [64]. Conversely, a study in human melanoma, non-small-cell lung cancer, small-cell lung cancer, epidermoid carcinoma, and ovarian cancer cells with an MDR phenotype (as a result of selection by drug exposure or by transfection with the *mdr1* gene) showed an increased sensitivity to gemcitabine. A potential explanation of these results is that P-gp and MRP1 overexpression caused cellular stress resulting in the increased activity of dCK, which catalyzes a rate-limiting reaction promoting gemcitabine metabolism and activity [65].

However, a study on the role of the GAIP-interacting protein C terminus (GIPC), which regulates the trafficking of endocytic vesicles, showed that when this factor is downregulated, there is metabolic intracellular stress which in turn causes autophagy. GIPC reduction also caused the increased expression of the *ABCG2* gene in the exosomes, which led to resistance to gemcitabine [66].

### 5.3. Modulation of Apoptosis Induction

Several preclinical studies have demonstrated the role of the impaired induction of apoptosis among the main causes of resistance to gemcitabine in PDAC, as reviewed previously [67]. These studies evaluated different mechanisms, such as TP53-dependent and independent pathways involved in gemcitabine-induced apoptosis, or the role of specific pro- or ant-apoptotic factors. In particular, Asuncion Valenzuela and collaborators [68] analyzed the inhibitors of apoptosis (IAP), a family of proteins including survivin, XIAP, cIAP1, and cIAP2, in order to understand their role in PDAC chemoresistance. Survivin, the smallest IAP, has a multifunctional role, including the regulation of mitosis, protection from cell death, and adaptation to stress conditions. This IAP is localized in the cytoplasm, mitochondria, and nucleus. However, there is also an extracellular pool of survivin which allows neighboring cancer cells to become resistant to therapy and rapidly proliferate, acquiring invasiveness [52]. Survivin, XIAP, cIAP1, and cIAP2 are upregulated by NF-KB in PDAC cells, and their overexpression correlates to resistance to chemotherapy. However, survivin can also be released by exosomes, which act as “multipurpose carriers” playing a role in the survival and growth of tumor cells, promoting immune response evasion, and facilitating metastasis. Extracellular survivin acts in the TME, causing resistance to therapy [52]. This study hypothesized that exosomal IAPs levels would reflect intracellular IAP levels, but the analyses showed only modest reductions in the exosomal IAP protein levels. This can be explained by a compensating effect of chemotherapy-treated cells to the decreased levels of IAPs in the exosomes, which in turn induces an increased release of exosomes into the extracellular space. Notably, exosomes also carry detectable levels of IAP mRNA, suggesting that both exosomal IAP mRNA and protein should be investigated for a better understanding of drug resistance mechanisms in PDAC. 

The potential role of exosomal survivin was also reported in a study using exosomes collected from engineered melanoma cells in order to contain the dominant negative mutant of survivin (Survivin-T34A) [69]. When these exosomes were plated on cancer cells, alone or in combination with gemcitabine, the authors observed a significant increase in apoptotic cell death when compared to gemcitabine alone on a variety of PDAC cell lines. Of note, these results support the use of exosomes as a new delivery method for anticancer proteins to target PDAC cells.

### 5.4. Modulation of Glutamine Metabolism and Reactive Oxygen Species

PDAC cells are characterized by several metabolic aberrancies, which impact on chemoresistance [70]. A recent study showed that glutaminase inhibitors sensitized chemoresistant PDAC cells to gemcitabine [71]. In keeping with these findings, a study on the modulation of a glutamine metabolic pathway to sensitize gemcitabine-resistant (GEM-R) pancreatic cancer cells, showed a reduction in endoplasmic reticulum (ER) proteins in the proteomes of the exosomes derived from GEM-R MiaPaCa cells treated with 6-diazo- 5-oxo-L-norleucine (DON) [72]. DON is a glutamine analogous that interferes with nucleotide and protein synthetic pathways. When disrupted, the glycosylation process can change the ER glycoprotein quality control by (1) activating the ER stress, (2) inhibiting protein synthesis, and (3) activating protein degradation. The increased presence of EGFR in the exosomes of the treated cells suggests a redirection of these proteins in the extracellular region, owing to glycosylation modulation. Consequently, the overexpression of the EGFR pathway confers chemoresistance. On the contrary, the downregulation of these proteins by damaging the glycosylation process should enhance the chemosensitivity to gemcitabine.

In addition, exosomes can induce chemoresistance with a mechanism that involves reactive oxygen species (ROS). Indeed, through superoxide dismutase 2 (SOD2) and catalase (CAT), exosomes increase the detoxification of ROS, and therefore protect the cells from the anticancer drugs [73]. Similarly, a study investigating the role of miR-155 in chemoresistance showed that exosomes are carriers of mRNA and miRNA modulating CAT, SOD2, and dCK and are therefore involved in ROS detoxification and gemcitabine metabolism [73].

### 5.5. Modulation of Oncogenic Pathways by Exosomal miRNAs

miRNAs have emerged as biomarkers for cancer prognosis and chemoresistance, and blood-based miRNAs are being evaluated as indicators of therapeutic activity in many tumor types, including PDAC. The up- or downregulation of miRNAs can affect the expression of multiple target mRNAs and proteins, leading to variations in the sensitivity of tumor cells via a number of different cellular processes. In particular, several miRNAs have been demonstrated to alter cellular response to anticancer agents via the modulation of drug efflux and targets, cell cycle, survival pathways, and/or apoptotic response with a key role in PDAC (Appendix A), as reported in previous reviews [74,75].

Here we describe miRNA affecting gemcitabine resistance which have been detected in exosomes and modulate key oncogenic pathways in PDAC.

#### 5.5.1. miR-155

Several studies support the role of exosomal miR-155 in drug resistance [52,73,76]. This miRNA is indeed overexpressed in PDAC after prolonged gemcitabine exposure and leads to a positive feedback loop which enhances gemcitabine resistance through increased exosomal release and the activation of antiapoptotic pathways. Using this mechanism, exosomes can also spread the drug resistance phenotype to the neighboring PDAC cells. A secondary mechanism underlying the role of miR-155 consists of targeting 3’UTR in dCK transcripts reducing the levels of the key protein in gemcitabine metabolism [77,78]. Of note, miR-155 is up-regulated in invasive IPMNs compared to non-invasive IPMNs, as well as in non-invasive IPMNs compared to normal pancreatic tissues [79]. Thus, tissue and exosomal miR-155 expression should be further investigated both as a potential diagnostic tool and as a predictive biomarker of chemoresistance.

#### 5.5.2. miR-210

Recent findings have highlighted the role of exosomes derived from gemcitabine-resistant PDAC stem cells, which can deliver chemoresistance traits to gemcitabine-sensitive stem cells: exosomes mediate this process by horizontal transfers of miR-210 which induces chemoresistance by activating the protein kinase B/mammalian target of rapamycin (AKT/mTOR) signaling pathway [80]. Other studies have shown that this miRNA is also involved in epithelial-mesenchymal transition (EMT) via NF-KB signaling pathway activation [81]. Both of these pathways are involved in cell survival and proliferation, suggesting that miR-210 blockade may be a promising therapeutic option in pancreatic cancer.

#### 5.5.3. miR-146a

It has been demonstrated that miR-146 and Snail were released through exosomes by pancreatic CAFs after exposure to gemcitabine. Once released, Snail and miR-146a are absorbed by epithelial cells, and can promote chemoresistance [78,82,83,84]. Against this background, a specific exosome blockade may be a promising therapeutic strategy for patients receiving gemcitabine-based treatments. Indeed, the in vitro suppression of the exosome release by CAFs reduces Snail expression, affecting the survival of resistant cells [83].

#### 5.5.4. miR-21

The oncogenic role of mR-21 n PDAC cell lines has been correlated with lower Phosphatase and TENsin homolog levels (PTEN) and increased mRNA expression of matrix metalloprotease 2 (MMP-2) that has also been involved in gemcitabine-induced apoptosis [85,86,87]. In addition, the decreased PTEN level decreased was associated with AKT upregulation and miR-21 levels were increased in the tumor cells of gemcitabine-resistant PDAC patients [88]. However, miR-21 has also been found to be highly released by pancreatic stellate cells and was associated with cell migration, EMT, and poor prognosis [85]. Moreover, another study investigated the role of miR-21 expression in CAFs which was involved in desmoplasia, gemcitabine chemoresistance in a PDAC xenograft model [87]. 

These findings suggest the potential role of miR-21 in different cells in the PDAC TME and recent studies have underlined the role of exosome-encapsulated miR-21 as sensitive marker in PDAC [89,90,91], suggesting that exosomal miR-21 might be a promising biomarker to assess the gemcitabine chemoresistance in PDAC patients.

**Table 1 diagnostics-12-00286-t001:** Exosome-mediated mechanisms of resistance to gemcitabine.

Exosome’s Content Involved in Resistance	Targets/Mechanisms Underlying Resistance	Cancer Model	References
GIPC	*ABCG2* gene	In vitro *MiaPaCa and BxPC-3*	[66]
miR-155	3’UTR of deoxycytidine kinase transcripts	In vitro	[77,78]
miR-210	AKT/mTOR	In vitro/in vivo *BxPC-3, PANC-1*	[80]
Snail miR-146	absorbed by epithelial cells, promotes chemoresistance	In vitro	[78,82,83,84]
EphA2	Transfers chemoresistance	In vitro *MIA PaCa-2 and BxPC-3, PANC-1*	[82,92,93]
miR-155	*CAT* gene *SOD2* gene *DCK* gene	In vitro	[73]
Nf-kb	Survin XIAP cIAP1 cIAP2	In vitro/in vivo *PANC-1*	[68]

## 6. Role of Exosomes in Resistance to Nab-Paclitaxel in Pancreatic Cancer

The resistance to nab-paclitaxel has been evaluated in a few studies and no data are available on mechanisms mediated by exosomes, but several studies have been performed to evaluate the mechanism of resistance to paclitaxel in different cancer types, and we speculate that similar mechanisms could play a role for nab-paclitaxel, as summarized in Figure 3.

### 6.1. Modulation of Multidrug Resistance Proteins

Cancer cells can counteract the accumulation of chemotherapeutic accumulation via drug efflux proteins such as the ATP-binding cassette (ABC) transporter family. A seminal study showed that inhibition of one of the protains of this family, P-gp, allowed paclitaxel biodistribution and efficacy to increase in an orthotopic brain tumor model [94]. 

Interestingly, drug efflux protein-mediated MDR can also be promoted by exosomes. For instance, P-gp can be directly transferred from resistant to sensitive cells via exosomes which promote an MDR phenotype [95]. Exosomes are also involved in the transfer of proteins and miRNAs that upregulate the P-gp expression from the donor to the recipient cell [96,97].

In order to assess the mechanisms underlying chemoresistance in nasopharyngeal carcinoma (NPC), which is a common cause of a low survival rate in these patients, the expression of DDX53 in taxol-resistant NPC cells was evaluated [98]. DDX53 was overexpressed in paclitaxel-resistant NPC cells and could also be transferred to other paclitaxel-sensitive cells through the secretion of exosomes. DDX53 acts by increasing the expression of MDR1, which enhances the resistance of NPC cells to paclitaxel.

### 6.2. Modulation of Apoptosis Induction

A study aimed at assessing the function of exosomes and their ability to transfer their contents in the context of breast cancer led to the identification of specific classes of exosomes enriched with survivin from aggressive cells treated with paclitaxel [99]. This protein can directly inhibit apoptosis by physically interacting with caspase, or indirectly via blocking of the mitochondrial pathway, and was regarded as a potential marker of paclitaxel resistance in breast cancer cells. 

### 6.3. Modulation of Oncogenic Pathways by Exosomal miRNAs

No data are available on miRNA affecting nab-paclitaxel, but several miRNAs have been associated with resistance to paclitaxel in different cancer types. For instance, dysregulation of the exosomal miRNA miR-433 [100] resulted in altered modulation of several signaling pathways leading to cell senescence and resistance to paclitaxel. Similarly, a study on the role of exosomes in prostate cancer evaluated 29 differently expressed exosomal miRNAs and showed 10 important pathways downregulated by exosomal miRNAs in prostate cancer cells [101]. Other pivotal exosomal miRNAs involved in resistance to paclitaxel are described in the following paragraphs.

#### 6.3.1. miR-21

One of the first studies focusing on the role of exosomes in chemoresistance to paclitaxel in ovarian cancer showed that exosomal miR-21 can confer chemoresistance to neighboring stromal cells [102]. The direct target of miR21 was APAF1, suggesting that an increase in APAF1 in ovarian cancer cells can be used to sensitize these cells to paclitaxel. In addition, it was seen that miR-21-5p was involved in detoxification mechanisms, while miR-21-3p and miR-891-5p were involved in DNA repair processes, demonstrating that these exosomal miRNAs could be implicated in resistance mechanisms to DNA-damaging agents in ovarian cancer cells [103]. 

#### 6.3.2. miR-155

Back in 2018, a preclinical study focused on the role of exosomal miR-155-5p in turning gastric cancer resistant to paclitaxel [104]. The authors created a paclitaxel-resistant gastric carcinoma cell line (i.e., MGC-803R), from which exosomes were extracted and used on paclitaxel-sensitive cells (i.e., MGC-803S) that acquired resistance. Further molecular studies showed that the targets that were affected by exosomal miR-155-5p were GATA binding protein 3 (GATA3) and tumor-inducible nuclear protein p53 (TP53INP1).

#### 6.3.3. miR-522

The most recent study [105] focusing on the role of ferroptosis in the induction of chemoresistance in gastric cancer assessed the exosomal secretion of miR-522 from CAFs. CAF-induced exosomal miR-522 secretion inhibited ferroptosis in gastric cancer cells by targeting arachidonate lipoxygenase 15 (ALOX15) which in turn blocked lipid-ROS accumulation within the cells. Paclitaxel promoted the secretion of miR-522 from CAF by suppressing ALOX15, thus decreasing lipid-ROS accumulation and finally decreasing chemosensitivity. In addition, paclitaxel promoted the secretion of exo-miR-522 from CAF by activating the ubiquitin-specific protease 7 (USP7)/heterogeneous nuclear ribonucleotin A1(hnRNPA1) axis.

#### 6.3.4. miR-1246

The role of exosomal miR-1246 was highlighted in a study assessing the role of exosomes in ovarian cancer resistance to paclitaxel [106]. The direct target of miR-1246 involved in the process of exosomal-mediated resistance transmission was the Cav1 gene. Indeed, patients with high exosomal miR-1246 and low Cav1 expression had a worse prognosis. The expression of miR-1246 was significantly higher in the exosomes of paclitaxel-resistant ovarian carcinoma cells than in paclitaxel-sensitive ones. Therefore, the use of miR-1246 inhibitors in combination with cytotoxic compounds was proposed as a therapeutic strategy to overcome paclitaxel chemoresistance in patients with ovarian cancer.

## 7. Discussion and Future Perspectives

The above-reported findings clearly show that exosomes are implicated in multiple mechanisms that might lead to chemoresistance, such as the release of molecules affecting apoptosis, detoxification, cellular metabolism and oncogenic pathways. This pleiotropic function supports further studies on the potential clinical impact of exosomes in the context of new therapeutic strategies to overcome drug resistance. The identification of new specific targets among the various molecules carried by exosomes can indeed guide new treatments to bypass the mechanisms of chemoresistance. 

Most studies have been focused on the exosomal transport of miRNAs as delivery vehicles for non-coding RNA-based cancer therapy, as recently reviewed [92], or on targeting pivotal PDAC oncogenes such as *KRas*. In particular, an ongoing clinical trial is using engineered exosomes (iExosomes) with the ability to target oncogenic Kras [107]. More recently, a proof-of-principle study demonstrated that exosomes loaded with CRISPR/Cas9 can also target the mutant *Kras G12D* oncogenic allele in PDAC cells in subcutaneous and orthotopic models [93].

However, an example of a protein target could be represented by exosomal Ephrin type-A receptor 2 (EphA2) which has recently emerged as a promising new target for novel anticancer drugs [108]. Exosomes isolated from PANC-1–chemoresistant cells can increase gemcitabine resistance to the “less resistant” MIA PaCa-2 and BxPC-3 cells through the transfer of EphA2 [82,109]. Interestingly, only exosomal EphA2 could transfer resistance while the soluble EphA2 did not promote chemoresistance. Even if EphA2 expression has not been yet validated in clinical trials, its potential downregulation has been investigated resulting in a lower cancer cell proliferation rate [109]. Moreover, the EphA2 inhibitor ALW-41-27 has already been tested in PDAC cells, showing its ability to reduce cell growth and migration [110]. Thus, exosomal EphA2 could be further studied as a potential marker for drug response and at the same time would represent a potential drug target.

In conclusion, preclinical data and emerging clinical evidence support further studies to validate the role of exosomes as predictive biomarkers of chemoresistance to specific anticancer drugs such as gemcitabine and nab-paclitaxel. Hopefully, in the near future the improved knowledge of the molecular mechanisms underlying such resistance will be used to improve the clinical management of PDAC patients and help the pharmacologists, oncologists and surgeons in the selection of more effective therapeutic strategies.

## Figures and Tables

**Figure 1 diagnostics-12-00286-f001:**
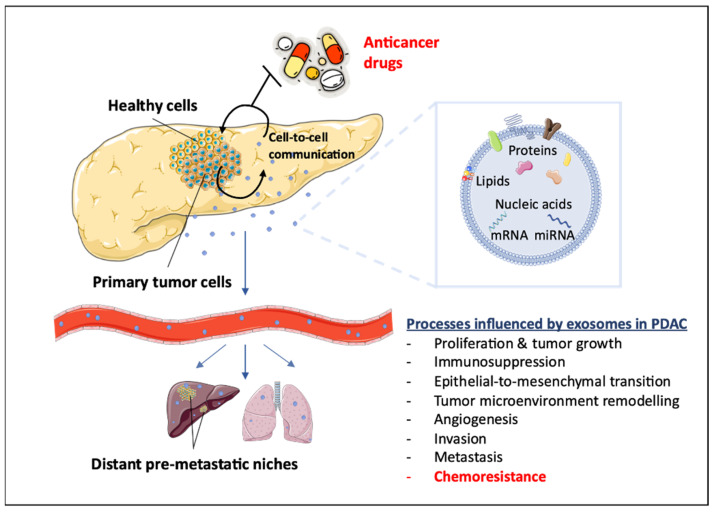
Schematic representation of exosome-related tumor processes. Exosomes mediate several important processes in PDAC pathogenesis, such as EMT and metastasis, and play a key role in pancreatic carcinoma chemoresistance.

**Figure 2 diagnostics-12-00286-f002:**
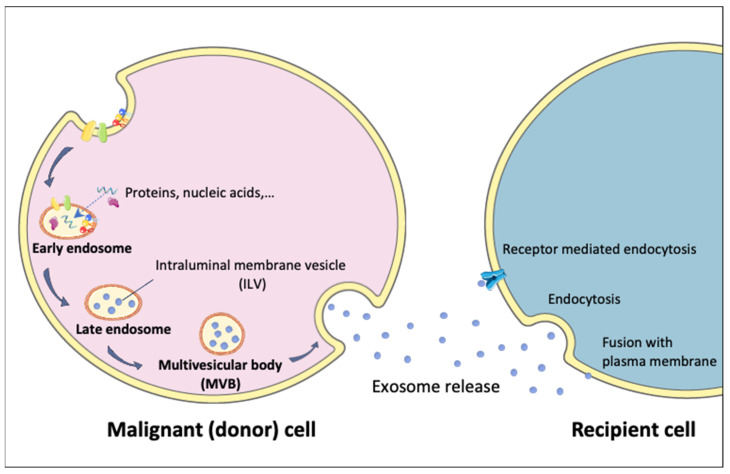
Schematic representation of exosome biogenesis and uptake. Exosome biogenesis proceeds in several steps: early endosomes are produced directly from the plasma membrane; ILV are included into the MVB formed after the maturation step involving a decrease in the pH; MVB fusion with plasma membrane led to exosome secretion; exosome capture by the recipient cell can be mediated via three mechanisms: (1) endocytosis by the plasma cell; (2) receptor-mediated endocytosis; (3) direct fusion with the plasma membrane.

**Figure 3 diagnostics-12-00286-f003:**
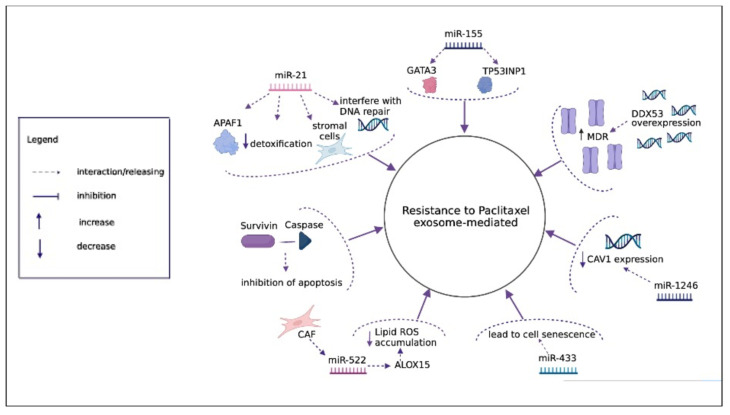
Exosome-mediated mechanisms of resistance to paclitaxel. The overexpression of DDX53 secreted by exosomes enhances the expression of MDR1. Exososomal survivin inhibits apoptosis interacting with downstream caspases. Exosomal miR-433 interferes with different pathways leading to cell senescence. Exosomal miR-21 confers resistance in different ways: targeting APAF1, being involved in detoxification and in DNA repair mechanisms. Exosomal miR-155 confers chemoresistance targeting GATA3 and TP53INP1. Exosomal miR-522 secreted from CAF suppresses ALOX15 that blocks lipid ROS accumulation. Exosomal miR-1246 mediate chemoresistance transmission targeting *Cav1* gene.

## Data Availability

Not applicable.

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
