# Peer review of "Potential Role of Exosomes in the Chemoresistance to Gemcitabine and Nab-Paclitaxel in Pancreatic Cancer"

_diagnostics, 2022, doi:10.3390/diagnostics12020286_

Round 1

Reviewer 1 Report

The review article by Comandatore et al. contains essential information regarding the role of exosomes in pancreatic ductal adenocarcinoma cancer (PDAC). 
The Authors provided the profound characteristic of the potential application of exosomes in the resistance to gemcitabine and nab-paclitaxel, which are two of the most commonly used drugs for the treatment of PDAC patients.
This review is well structured, with fine paragraphs concerning the most current knowledge of exosome-related tumour processes. The role of exosomes in different mechanisms that might lead to chemoresistance and can be related to the release of molecules affecting apoptosis, detoxification, cellular metabolism and an oncogenic pathway was profoundly characterized and described in detail.  The Authors also paid attention to pancreatic tumour resistance, but also the mechanisms underlying resistance to the gemcitabine plus nab-paclitaxel regimen. The Authors provided an impressive review paper, that is well written and planed. I really like the  „Discussion and future perspectives” chapter which highlight the importance of the knowledge discussed in this paper to improve the clinical management of PDAC patients. 
This article will be of great interest to the readers of Diagnostics.

Author Response

We are grateful to the reviewer for his/her careful evaluation of the manuscript and the positive comments.

Reviewer 2 Report

The manuscript "Potential role of exosomes in the chemoresistance to gemcita-2 bine and nab-paclitaxel in pancreatic cancer.“ is up to date. It is well written with information about exosomes and their role in chemoresistance of pancreatic cancer cells. I'm just missing the little things:

  1. I suggest you to make table where you summarise all posible biomarkers fto investige pancreatic cancer progresion and also the biomarkers for prediction of sensitivity of PDAC patients to chemotherapy. It will be more informative for the readers.

After incorporation of this comment I suggest to publish this review manuscript.

Author Response

We appreciate very much the constructive comments of this Reviewer. In agreement with his/her suggestion we have included in the revised article a table (Supplementary Table 1) on “Candidate pancreatic cancer biomarkers and their potential application.”